# Comparative Analysis of Transcriptomes of *Ophiostoma novo-ulmi* ssp. *americana* Colonizing Resistant or Sensitive Genotypes of American Elm

**DOI:** 10.3390/jof8060637

**Published:** 2022-06-16

**Authors:** Martha Nigg, Thais C. de Oliveira, Jorge L. Sarmiento-Villamil, Paul Y. de la Bastide, Will E. Hintz, Sherif M. Sherif, Mukund Shukla, Louis Bernier, Praveen K. Saxena

**Affiliations:** 1Centre d’Étude de la Forêt (CEF) and Institut de Biologie Intégrative et des Systèmes (IBIS), Université Laval, Québec, QC G1V 0A6, Canada; martha.nigg@gmail.com (M.N.); thais.campos-de-oliveira.1@ulaval.ca (T.C.d.O.); jorge-luis.sarmiento-villamil.1@ulaval.ca (J.L.S.-V.); 2Department of Biology, Centre for Forest Biology, University of Victoria, Victoria, BC V8W 2Y2, Canada; pdelabas@uvic.ca (P.Y.d.l.B.); whintz@uvic.ca (W.E.H.); 3Alson H. Smith Jr. Agricultural Research and Extension Center, School of Plant and Environmental Sciences, Virginia Tech, Winchester, VA 22602, USA; ssherif@vt.edu; 4Department of Plant Agriculture, Gosling Research Institute for Plant Preservation (GRIPP), University of Guelph, Guelph, ON N1G 2W1, Canada; mshukla@uoguelph.ca

**Keywords:** Dutch elm disease, *Ophiostoma novo-ulmi*, RNA-Seq, pathogenicity, virulence, mutants

## Abstract

The Ascomycete *Ophiostoma novo-ulmi* threatens elm populations worldwide. The molecular mechanisms underlying its pathogenicity and virulence are still largely uncharacterized. As part of a collaborative study of the *O. novo-ulmi*-elm interactome, we analyzed the *O. novo-ulmi* ssp. *americana* transcriptomes obtained by deep sequencing of messenger RNAs recovered from *Ulmus americana* saplings from one resistant (Valley Forge, VF) and one susceptible (S) elm genotypes at 0 and 96 h post-inoculation (hpi). Transcripts were identified for 6424 of the 8640 protein-coding genes annotated in the *O. novo-ulmi* nuclear genome. A total of 1439 genes expressed in planta had orthologs in the PHI-base curated database of genes involved in host-pathogen interactions, whereas 472 genes were considered differentially expressed (DEG) in S elms (370 genes) and VF elms (102 genes) at 96 hpi. Gene ontology (GO) terms for processes and activities associated with transport and transmembrane transport accounted for half (27/55) of GO terms that were significantly enriched in fungal genes upregulated in S elms, whereas the 22 GO terms enriched in genes overexpressed in VF elms included nine GO terms associated with metabolism, catabolism and transport of carbohydrates. Weighted gene co-expression network analysis identified three modules that were significantly associated with higher gene expression in S elms. The three modules accounted for 727 genes expressed in planta and included 103 DEGs upregulated in S elms. Knockdown- and knockout mutants were obtained for eight *O. novo-ulmi* genes. Although mutants remained virulent towards *U. americana* saplings, we identified a large repertoire of additional candidate *O. novo-ulmi* pathogenicity genes for functional validation by loss-of-function approaches.

## 1. Introduction

*Ophiostoma novo-ulmi* is the highly aggressive ascomycete responsible for the ongoing Dutch elm disease (DED) pandemic. Within a few decades, it has taken over the less aggressive *O. ulmi* which initiated the first pandemic of DED a little more than 100 years ago [1]. Within *O. novo-ulmi*, two subspecies have been identified, *novo-ulmi* and *americana* [2]. Both subspecies are found in Europe, as well as in Eastern and Central Asia [3]. Subspecies *americana* is also present in North America, Japan, and New Zealand [3,4,5]. A third species, *O. himal-ulmi*, was discovered in symptomless elms in the western Himalayas and shown to be pathogenic to European elms by controlled inoculations [6]. The first full genome sequences of DED fungi were released in 2013. They were obtained for *O. ulmi* strain W9 [7] and *O. novo-ulmi* ssp. *novo-ulmi* strain H327 [8]. The latter is a model for investigations of DED pathogens [9,10] and its genome has been fully assembled and annotated [8,11]. This, in turn, has facilitated assemblies of additional genomes used in population studies of *O. ulmi* and *O. novo-ulmi* [12].

In nature, DED fungi are vectored by elm bark beetles belonging to the genera *Scolytus* and *Hylurgopinus* [13]. Young adult bark beetles feed in the crown of adult, healthy elms in early summer. Beetles carrying conidia of DED fungi on their exoskeleton inoculate the pathogen when they bore though the bark of twig crotches. Once inoculated, the fungus grows into the water conducting system and induces tree defense responses. The DED fungi are dimorphic (i.e., capable of switching growth phases from yeast to mycelium in a reversible manner, depending on environmental conditions) [14]. Expressed sequence tag (EST) analyses of *O. novo-ulmi* strain H327 during yeast growth allowed the identification of close to 2100 genes expressed under this growth form [15]. Subsequently, RNA-Seq-based transcriptomic studies of strain H327 grown in vitro showed that 12% of the gene content was differentially expressed between the two growth phases [16] and that the transcriptome changed significantly within two hours of the transition from yeast growth to mycelial growth [17]. Although yeast-mycelium transition is thought to contribute to parasitic fitness of the DED fungi [10], no in planta expression data are available for this trait.

The genes and molecular pathways that allow DED fungi to invade host tissues following inoculation are largely unknown. Analysis of F_1_ progeny from controlled crosses between *O. ulmi* and *O. novo-ulmi* (originally thought to be a more aggressive form of *O. ulmi*) indicated that pathogenicity was under polygenic control [18]. Annotation of the *O. novo-ulmi* H327 genome revealed that 1731 genes (20% of gene content) had orthologs in the PHI-base curated database of genes involved in host-pathogen interactions [11]. Yet, very little experimental evidence is available for candidate pathogenicity genes of DED fungi. For instance, the *cu* gene (OnuG4296) encoding a hydrophobin known as cerato-ulmin [19] was shown to be important for spore attachment to bark beetles [20] but not for pathogenicity towards American elm saplings [21]. Likewise, gene *epg1* (OnuG1567) encoding an endopolygalacturonase is likely not a major contributor to pathogenicity since a null mutant was only slightly less virulent than its wild-type progenitor [22]. The *pat1* locus, identified by genetic analysis of crosses between *O. novo-ulmi* H327 and a less virulent introgressant [23], was mapped to a 40 Kb long portion of the largest chromosome of strain H327 genome [24] but it is not known whether it is directly controlling pathogenicity or tightly linked to a *bona fide* pathogenicity gene. Mapping data and functional characterization are also required for a yet uncharacterized gene involved in the production of yeast-like spores and synnematal spores, and possibly pathogenicity [25,26].

Functional validation of candidate pathogenicity genes in *O. novo-ulmi* has long been impeded by the low recovery frequency of targeted mutants, which makes the screening of potential mutants a tedious procedure [22]. The impact of this obstacle was lowered by using RNA interference [27] to produce knockdown mutants containing a carbon source regulated promoter to drive cassette expression and downregulate specific genes or gene families identified through sequence analysis [28,29]. More recently, targeted knockout mutants were recovered at high frequency after subjecting a Δ*mus52* derivative of strain H327 defective for non-homologous end-joining of DNA (NHEJ) to a dual selection system in OSCAR plasmids [30]. The next development in this area will likely be the development of protocols based on CRISPR/Cas (Clustered regularly interspersed short palindromic repeats/CRISPR associated; [31]) for precise genome editing in *O. novo-ulmi* [32]. Functional validation of candidate pathogenicity genes would also be facilitated by data from comprehensive transcriptomic analyses of elm-*O. novo-ulmi* interactions. Previous research, however, has focused on transcripts from the host [33,34] rather than from the pathogen. One technical reason for this situation has been the difficulty in recovering sufficient amounts of fungal transcripts since the latter represent a small proportion of the total population of messenger RNAs in inoculated elm tissue. This is no longer a problem thanks to continuing improvements in high throughput sequencing. Here we present the transcriptomes of *O. novo-ulmi* ssp. *americana* strain MH75-4O recovered from susceptible as well as resistant American elms and subjected to deep RNA sequencing. Elm transcriptomes obtained from the same experiment were recently presented in a distinct publication [35]. We conducted in silico analyses of the fungal transcriptomes and assessed the contribution of selected genes to pathogenicity by inoculating knockdown- and knockout mutants of *O. novo-ulmi* ssp. *novo-ulmi*.

## 2. Materials and Methods

### 2.1. Fungal Isolates

Due to quarantine issues, elm saplings used for the transcriptome analysis conducted at the University of Guelph campus (Guelph, ON, Canada) could not be inoculated with reference strain *O. novo-ulmi* ssp *novo-ulmi* H327. Instead, elms were inoculated with *O. novo-ulmi* ssp. *americana* strain MH75-4O which had been used in previous molecular investigations of American elm response to infection [36]. Strain MH75-4O originates from strain MH75 collected in Toronto (ON, Canada) by Martin Hubbes and has since been sequentially inoculated and recovered four times. Mutant strains for functional validation of candidate genes were derived from strain *O. novo-ulmi* H327. Controlled inoculations of mutants were carried out in facilities on the Laval U. campus (Quebec, QC, Canada) that were certified PPC-1 by the Canadian Food Inspection Agency. All strains are kept at −80 °C in the Centre for Forest Research (Quebec, QC, Canada) culture collection.

### 2.2. Inoculations and Recovery of Transcripts from Elm-Ophiostoma novo-ulmi Interaction

Four-year-old saplings of two types of American elm (*Ulmus americana*) were used for transcriptomic analysis of the elm-*O. novo-ulmi* interaction: DED resistant ‘Valley Forge’ (VF) purchased from Connon Nurseries (West Flamborough, ON, Canada) and susceptible (S) elm plant material selected from clones of a susceptible elm tree in the in vitro germplasm bank at the Gosling Research Institute for Plant Preservation (GRIPP), University of Guelph. Saplings were inoculated with strain MH75-4O. Inoculations were carried out in triplicate. Total RNA was extracted from inoculated elms at 0 and 96 h post inoculation (hpi), respectively, as described previously [35,36]. Twelve complementary DNA (cDNA) libraries were subjected to RNA-Seq. Detailed procedures for inoculation, RNA extraction, RNA sequencing and filtering of raw RNA-Seq data are provided in Islam et al. [35].

### 2.3. Ophiostoma novo-ulmi ssp. americana Transcriptome Mapping and Analysis

Filtered reads were mapped onto the *O. novo-ulmi* ssp. *novo-ulmi* H327 exonic sequences [8] with TopHat2 (v.2.0.10) using default parameters for paired-end RNA-Seq sequencing products [37]. All further analyses were performed using R (v3.0.1; [38]). Read counts for each of the twelve samples were extracted and used for differential expression analyses using the *EdgeR* package in Bioconductor 3.0 [39]. Genes that were represented by a total of at least three reads (cpm > 3) among the three replicates of a condition were kept for downstream analyses. Library sizes (i.e., number of mapped reads) were corrected using a method based on Trimmed Mean of M-values (TMM) [40,41].

A gene was considered upregulated when it was over-expressed at least with a log_2_ Fold Change (log_2_FC) = 1 (thus expressed at least two times more in one of the two conditions). Gene Ontology (GO) term enrichment analyses were performed on sets of overexpressed fungal genes at 96h in S and VF varieties as described previously [16,17]. *Ophiostoma novo-ulmi* gene transcription patterns during infection of S and VF *U. americana* saplings were investigated by Weighted Gene Co-expression Network Analysis (WGCNA) using the WGCNA package in R [42]. The gene matrix was constructed with a threshold of 20 for *n* = 6. All modules were hierarchically clustered based on topological overlap matrix (TOM) similarity. Modules containing 100 genes or less were merged with their closest larger neighbor module. Representative Kyoto Encyclopedia of Genes and Genomes (KEGG; [43]) pathways were identified within modules that were significantly associated with different levels of fungal expression in S and VF elms. Interaction networks of genes of interest within modules were visualized using Cytoscape 3.9.0 [44].

### 2.4. Functional Analysis of Candidate Pathogenicity Genes

Nine candidate *O. novo-ulmi* genes were selected for functional analysis based on their expression in planta or in vitro (see 3.4 for details). We first produced knockdown mutants by RNA interference of the following genes: *amtA* (OnuG0282), *mad1* (OnuG3773), *hex1* (OnuG6790.1), *cyp570* (OnuG7411), *cyp52P6* (OnuG7466) and *cox2* (OnuG5504.1). RNAi cassettes were built to down-regulate the expression of candidate genes using *O. novo-ulmi* H327 nucleotide sequence data [11]. A DNA cassette was constructed to drive the expression of a gene-specific double-stranded RNA (dsRNA) based upon a functional coding region of each target gene. The basic design of the RNAi cassette was synthesized by GeneArt Gene Synthesis (Invitrogen, ThermoFisher Scientific, Mississauga, ON Canada), inserted into the holding vector pMK-RQ (kanR) at cloning sites of SfiI/SfiI and purified after growth in the transformed bacterial host *Escherichia coli* K12 DH10B T1R. Each RNAi construct included a initial *Spe*I restriction enzyme site, an approximately 600 bp antisense sequence followed by an approximately 400 bp sense sequence and a terminal *Kpn*I restriction enzyme site to facilitate insertion into the transformation vector pAN7-1 (Appendix A).

A universal expression cassette was designed to evaluate candidate genes using the pAN7-1 fungal transformation vector and was synthesized by GeneArt Gene Synthesis. The *Trp*C transcription terminator of the pAN7 vector [45] was excised by a *Bam*HI/*Hind*III restriction digest and replaced by the *O. novo-ulmi* alcohol dehydrogenase (*alc*A) promoter that was used to regulate RNAi cassette expression, as it is differentially regulated by carbon source [29]. A version of the *alc*A promoter lacking two *Hind*III restriction sites was synthesized to facilitate fungal transformations using an existing *Hind*III site in the pAN7-1 vector. The transcription of RNAi cassettes inserted at the multiple cloning site (MCS) was terminated by the *Trp*C terminator from *Aspergillus nidulans* [45]. The screening of holding vectors and candidate gene sequences identified six restriction enzymes suitable for the MCS (*Spe*I, *Bsp*DI, *Asc*I, 8 nucleotide spacer (ATGCGAAG), *Xma*I, *Not*I and *Kpn*I) that was placed between the *alc*A promoter and *Trp*C terminator to facilitate the insertion of each RNAi cassette tailed with unique restriction sites (*Trp*C-*alc*A-(*Spe*I)-RNAi cassette-(*Kpn*I)-*Trp*C).

The pAN7 vector includes a constitutive glyceraldehyde-3-phosphate dehydrogenase (gpd) promoter sequence of *A. nidulans* [45] that drives expression of the hygromycin resistance gene (*hph*) and is tailed by endogenous *Bam*HI and *Hind*III restriction sites. Expression of *hph* confers hygromycin resistance and confirms successful integration of transforming DNA. Insertion of each RNAi cassette was facilitated by a *Spe*I/*Kpn*I restriction enzyme double digest in which the pAN7 vector loses 46 bp within the MCS and the pMK vector releases an approximately 1000 bp fragment containing the RNAi cassette. All constructs were verified by nucleotide sequence analysis to confirm the presence of a functional cassette, prior to their use in transformations (Appendix A).

Protoplasts were prepared [20] and transformed [28] as described previously. Colonies growing on Ophiostoma Complete Medium (OCM) agar [46] supplemented with 300 μg/mL hygromycin were transferred to liquid OCM (300 μg/mL hygromycin B) containing glycerol as sole carbon source rather than glucose to activate the *alc*A promoter and promote RNAi cassette expression; these cultures were used for subsequent RNA extraction protocols to evaluate gene expression. Suspected positive transformants were screened by conventional PCR using primer annealing sites located in the transformation vector sequence just outside of the MCS; positive transformants yielded a 1550 to 1650 bp fragment, depending upon RNAi cassette size. Transformants identified on selective medium and screened for successful integration of the RNAi cassette were selected for functional validation by quantitative PCR (qPCR) analysis. To assess target gene expression, levels of gene-specific messenger RNA (mRNA) transcribed in cultures of the H327 WT isolate were compared to the corresponding RNAi mutant by qPCR. Total RNA extracted from liquid OCM cultures [28] was used for complementary DNA (cDNA) synthesis with the Omniscript Reverse Transcription Kit using Oligo-dT Primers (Qiagen Inc.) to generate first strand cDNA from a sample of approximately 1 μg *O. novo-ulmi* RNA.

The qPCR reporter primers amplified a portion of the target gene locus using a forward primer site located in the coding region towards the end of the RNAi cassette sequence and a reverse primer site located on the complementary strand in the coding sequence outside of the RNAi region. This ensured that qPCR analysis of mutant and WT strains only measured and compared complete mRNA transcripts of expected size and melting point for the gene under study (Appendix A). The design of novel primers for the target loci was assisted by Primer3 (Version 4.1.0) (http://primer3.ut.ee (accessed on 12 July 2017)) and MEGA7 (Version 7.0.26) software. Reference sequences of the candidate genes from *O. novo-ulmi* H327 assisted primer design. To compensate for variability in cDNA quantity, endogenous reference genes of actin (*act*), β-tubulin (*Bt*) and chromosomes protein 1 (*psm1*) were designed with a desired amplicon length of 80 to 200 nucleotides and an annealing temperature of 58–60 °C (Appendix A). All primers were synthesized by Eurofins Genomics (Huntsville, AL, USA) and validated by testing a dilution series of cDNA (1:2, 1:10, 1:50, 1:250, 1:1250 template DNA) to create a standard curve based upon cDNA concentration and C_t_ values. An acceptable curve would have R^2^ > 0.98 and Efficiency from 90% to 110%.

To assess the influence of RNAi transformation on host gene expression, qPCR reaction mixtures were prepared in a 20 µL final volume that included Applied Biosystems Power SYBR Green Master Mix (ThermoFisher Scientific, Waltham, MA, USA), 0.05 µM each of the forward and reverse primers for target genes, control cDNA or reference genes (*act*, *Bt* and *psm1*), and 2 µL of template cDNA from sample material (3 replicates/sample). The reactions were run in a StepOnePlus Real-time PCR System (Applied Biosystems, Waltham, MA, USA, ThermoFisher Scientific), using a protocol that included an initial 10 min at 95 °C, followed by 35 cycles of 15 s at 95 °C and 35 s (80–99 bp amplicon), 40 s (100–130 bp amplicon), or 45 s (130–200 bp amplicon) at 59 °C, followed by an amplicon melt curve running up to 95 °C in 0.3 °C increments. Sample results for positive amplicons were scored by comparison to the reference T_m_ and subsequent data analyses were completed with the assistance of the StepOne software (Version 2.3). The products of qPCR amplification for target genes were initially characterized by nucleotide sequence analysis to confirm their identity. Samples that provided ambiguous results for melt curve analysis were also run by conventional PCR, followed by gel electrophoresis and visualization to confirm amplicon size.

Knockout mutants for genes *aox1* (OnuG5955), *opf2* (OnuG1642) and *bct2* (OnuG2340) were produced using *Agrobacterium tumefaciens*-mediated transformation (ATMT) of an *O. novo-ulmi* ∆*mus52* strain deficient in non-homologous end joining (NHEJ), followed by counterselection of ectopic integration of the T-DNA using binary vector pOSCAR-HSVtk [30,47]. Deletion plasmids were prepared as described previously [30,48]. Primers designed to amplify the region flanking the ORF of the target genes contained one of four different *att*B recombination sites in their 5′ extension (Appendix A). The two fragments amplified for each gene, pA-NTC-OSCAR [48] and pOSCAR-HSVtk were incubated for 20 h in the presence of BP clonase. Afterwards, One Shot OmniMAXTM 2 T1 *Escherichia coli* competent cells were transformed with each of the PB clonase reaction mixtures. Then, right deletion constructs were confirmed by restriction enzyme digestion. The deletion constructs were introduced into *A. tumefaciens* strain GV3101:pMP90 by the heat shock method, and ATMT of *O. novo ulmi* with deletion constructs was performed as described previously [49] with minor modifications. Transformed fungal cells were plated on Potato Dextrose Agar (PDA) supplemented with 50 µg/mL nourseothricin and 50 mg/mL 5-fluoro-20-deoxyuridine (5FU) to select transformants and counterselect ectopic transformants, respectively. Correct DNA integration was confirmed by PCR for individual transformants, using target gene-specific primer combinations Oao1F_236/Oao1R_840, Opf2R_254/Opf2R_873, and Bct2F_539/Bct2R_1111 (Appendix A), which amplified roughly 600-bp ORFs of *aox1*, *opf2*, and *bct2*, respectively.

Knockdown- and knockout mutants were phenotyped in vitro and in planta. Laboratory tests included measurements of mycelial growth rate at 21 °C on Malt Extract Agar (MEA) and MEA supplemented with 0.2 M NaCl (for *hex1* and *mad1* mutants) or 1:1000 limonene (for *cyp570* and *cyp52P6* mutants). Each assay was run on triplicate samples and growth rate over up to 10 days was recorded [50]. Pathogenicity was first assessed on Golden Delicious (GD) apples [51]. Necroses were measured at two- and four-weeks post-inoculation and each treatment (including a non-inoculated control) included at least eight biological replicates. Pathogenicity tests were also conducted on two-year-old *U. americana* saplings grown from seeds collected on the Laval University campus. Saplings were inoculated either in a greenhouse compartment or in a growth chamber kept at 18 °C during the night and 24 °C during the day, with a 16-h photoperiod and 60% humidity [52]. In addition to mutants for candidate pathogenicity genes, treatments included WT strain H327, deletion mutant Δmus52 (OnuG0757), and a negative control consisting in sterile distilled water. Each treatment was applied to 10 individual saplings in a randomized block design, using a procedure described previously [30]. Defoliation was recorded for each sapling two and three weeks after inoculation and the mean defoliation was calculated for each treatment. The performance of mutants compared to WT strain H327 (for RNAi mutants) or mutant strain Δmus52 (for KO mutants) was assessed by analysis of variance and post-hoc tests.

## 3. Results

### 3.1. Overview of the O. novo-ulmi ssp. americana Transcriptome in Planta

Since the genome of *O. novo-ulmi* ssp. *americana* strain MH75-4O is not sequenced, we instead used the sequence of *O. novo-ulmi* ssp. *novo-ulmi* strain H327 to map transcripts. As there is only 0.5% pairwise sequence divergence between these genomes [12], we expect them to be very similar.

The total numbers of reads (right and left reads taken together) at 0 hpi and 96 hpi in the three replicates of each condition were comprised between 66 million and 76 million (Table 1). However, less than 0.1% of reads mapped to the exonic sequences of the *O. novo-ulmi* H327 genome. A few reads from the 0 hpi treatments (40 on average) mapped partially to the *O. novo-ulmi* H327 genome but none mapped fully to *O. novo-ulmi* exons. These reads were considered to be artefacts or genes from fungal endophytes. At 96 hpi, the number of reads mapped to *O. novo-ulmi* exons was lower in resistant VF elm (*n* = 24 193, 0.033% of Total reads) than in S elm (*n* = 65 910, 0.087% of Total reads).

We compared the *O. novo-ulmi* gene expression levels at 96 hpi in S (96h_S) and VF (96h_VF) elms. After filtration, we kept 6424 genes (76% of the *O. novo-ulmi* H327 gene set) represented with at least three reads in one of the two conditions. We found 5392 genes expressed in 96h_S and 3014 in 96h_VF (Table 2). There was a strong correlation (R^2^ = 0.870; *p* < 0.0001) between *O. novo-ulmi* gene expression levels in 96h_S and 96h_VF treatments, i.e., most genes had a similar relative ranking in the two datasets. The top ten most expressed fungal genes in both conditions included orthologs of five heat shock protein (HSP)-encoding genes (OnuG2162, OnuG4489, OnuG7476, OnuG6249, and OnuG2380) (Appendix A). A subset of 303 genes expressed in 96h_S were not detected in 96h_VF elms, whereas only one gene (OnuG2030) expressed in 96h_VF (*n* = 1.7 transcripts) was not in 96h_S elms.

Out of the 6424 *O. novo-ulmi* genes expressed in planta, 1439 had orthologs in the PHI-base curated database of genes involved in host-pathogen interactions (Appendix A; www.phi-base.org, last accessed 18 April 2022; [53,54]). This number represents 83% of the *O. novo-ulmi* genes with orthologs in PHI-base [11]. The 1439 PHI-base *O. novo-ulmi* orthologs included nine genes which had not been detected in previous transcriptomic analyses of yeast cells and mycelium grown in vitro [15,16,17]. These genes were typically expressed at low levels (mean number of transcripts from 0.3 to 17.0 in one or the other of the two elm genotypes), except for OnuG5955 which was highly expressed in both 96h_S and 96h_VF (Appendix A). Genes from 15 of the 19 *O. novo-ulmi* secondary metabolite biosynthetic gene clusters identified in silico [55] were expressed in planta. When all fungal genes were considered, the *U. americana*-*O. novo-ulmi ssp. americana* transcriptome dataset included 90 genes that had not been detected in previous laboratory studies (Appendix A). Only 25 of these genes were represented by 10 or more transcripts in the 96h_S or 96h_VF datasets. Genes detected only in planta were located on all eight *O. novo-ulmi* chromosomes, although distribution of the genes in *the O. novo-ulmi* H327 genome was not random: there were 10 groups of two to four adjacent genes, as well as one group of 10 genes tightly linked on chromosome 2. This region, however, was not considered a cluster in the *O. novo-ulmi* genome annotation [11]. Conversely, 2538 genes expressed in vitro were not detected in planta, including 102 genes with orthologs in PHI-base.

### 3.2. Differentially Expressed Genes in O. novo-ulmi

We found 472 genes that were differentially expressed (DEG) between 96h_S and 96h_VF (Table 2 and Figure 1A). A subset of 370 genes were upregulated in 96h_S (Appendix A) and 102 genes were upregulated in 96h_VF (Appendix A). In general, the level of gene expression was comprised between 0 and 7000 reads per gene. We found 162 DEGs that were expressed in 96h_S but not in 96h_VF (Appendix A).

The 370 genes overexpressed in 96h_S included 94 genes listed in PHI-base (Appendix A). One third (34) of these genes were not expressed at 96h post inoculation in VF elm. We will further present the different categories of genes that were present among the 94 genes from PHI-base.

Three cytochrome P450 (CYP) encoding genes included in PHI-base were part of the five *cyp* genes overexpressed in 96h_S (Appendix A). One of these three *cyp* genes codes for an O-methylsterigmatocystin oxidoreductase (*cyp620T1*, OnuG6511). We also found the gene encoding a sterigmatocystin 8-O-methyltransferase (OnuG7303), which is not a CYP but is involved in the same molecular pathway. Despite the very low level of OnuG7303 expression in the S elm variety, the difference between the number of reads in 96h_S and 96h_VF was significant since the gene was not expressed in VF elm. The other two *cyp* genes included in PHI-base code for a benzoate 4-monooxygenase (*cyp5128E2*, OnuG7958) and a putative α-Pinene to Verbenol enzyme (*cyp52P6*, OnuG7466). Finally, the two *cyp* genes without orthologs in PHI-base code for another benzoate 4-monooxygenase (*cyp53E2,* OnuG3693) and a pisatin demethylase (*cyp570A4*, OnuG7411), respectively.

We identified four genes that are predicted to code for Polyketide Synthases (PKS) (OnuG1078, OnuG1434, OnuG6972 and OnuG7312). These genes were described as cluster/backbone genes and reannotated as *pks1*, *pks*2, *pks*7 and *pks*8, respectively [55]. Another gene upregulated in S elm, OnuG8024, is one of four *O. novo-ulmi* genes predicted to encode a putative HC-toxin efflux carrier TOXA. We also found one ortholog of the gene encoding TOXD (OnuG7311), involved in the metabolism of the HC-toxin in *C. carbonum*. This gene was not expressed in 96h_VF nor in datasets from previous laboratory studies. However, it was expressed at a very low level (4.3 transcripts) in the S variety.

There were 21 transporter-coding genes, including three genes coding for ammonium transporters (OnuG1681, OnuG4730 and OnuG0282). Gene OnuG1681 encoding an ortholog of the Mep2 transporter was found in the top 10 genes that were the most differentially expressed in 96h_S (log_2_FC = 3.21). Genes associated with nitrogen compound transport (GO:0071705) were the most overrepresented in the portion of genes overexpressed in 96h_S vs. 96h_VF (Figure 1B and Appendix A). In fact, GO terms for processes and activities associated with transport and transmembrane transport globally accounted for half (27/55) of the GO terms that were significantly enriched in fungal genes upregulated in S elms. The latter also included a subset of 17 GO terms associated with regulation of various biological processes.

None of the 13 genes predicted to code for secreted peptidases mentioned by Comeau et al. [11] as likely being associated with pathogenicity were found in the sets of fungal genes from S or VF inoculated elm saplings. Whereas the *cu* gene (OnuG4296) encoding the well described hydrophobin known as cerato-ulmin was not found to be DEG, it was clearly more expressed in the S elm variety (Appendix A). Various other candidate pathogenicity genes, including 14 genes coding for lipases and five effector-like encoding genes, showed the same expression pattern in both elm genotypes. Finally, we found seven genes encoding proteins with signal peptides that are in PHI-base, including six genes which were not expressed in the VF variety. Regardless of the very low expression level of these genes, they are candidates for further experiments focusing on fungal pathogenicity.

Genes linked with the metabolism, catabolism and transport of carbohydrates were overrepresented in the set of 102 fungal genes upregulated in 96h_VF compared to 96h_S (Figure 1B and Appendix A). The set included 17 CAZyme-encoding genes, consistent with an activation of the carbohydrate metabolism. Twelve predicted CAZymes were glycoside hydroxylases (GH), whereas 16 had a peptide signal. GO terms for aerobic (GO:0009060) and cellular (GO:0045333) respiration were also enriched, as well as terms associated with oxidoreductive processes (GO:0055114) and functions (GO:0016491; GO:0016702).

Compared with genes highlighted in vitro during the dimorphic switch in *O. novo-ulmi* [17], we found no evidence here of conservation of expression since most of the genes that were overexpressed in the hyphal growth phase (27 h post stimulus) versus the yeast phase were not highly expressed in planta. One notable exception was OnuG7491, encoding an ortholog of STE12 (PHI:2187), which was significantly more expressed both in yeast [16] and in 96h_S than in 96h_VF. Two additional genes, OnuG3117 (Aquaporin-1) and OnuG8465 (Possible guanine deaminase), significantly more expressed in 96h_S than in 96h_VF elms, were also differentially expressed in yeast and mycelium, respectively [16,17]. The two genes were part of a subset of 19 DEGs found to be under significant positive selection by Hessenauer et al. [12] who compared populations of DED fungi.

### 3.3. Gene Expression Network Analysis

Twenty-nine modules were identified and clustered by WGCNA of the 6424 genes expressed in planta (Figure 2A). Each module represents a cluster of highly interconnected *O. novo-ulmi* genes with similar expression changes during infection of elm. The number of genes within modules varied from 103 to 490 (Figure 2B). The mean number of transcripts for each hub gene (genes with the most connections within a module) was low (means of 19.4 and 7.3 in S and VF-elms, respectively). The notable exception was the hub gene for the Pink module (OnuG8227) encoding a heat shock protein HSP88 and for which 333 and 134 transcripts were detected in S and VF elms, respectively. Tree cluster hierarchy of gene expression modules was highly similar for the 96h_S and 96h_VF datasets. Module-trait relationships were significantly correlated for modules Darkred (*p* < 0.001), Green (*p* < 0.001) and Purple (*p* < 0.05) between 96h_S and 96h_VF datasets, with fungal gene transcript numbers higher in the susceptible elm genotype (Figure 2C). These modules contained 156, 318 and 253 genes, respectively, and included 28 genes encoding proteins with a signal peptide, 16 CAZY genes and 76 genes with an ortholog in PHI-base (Figure 3A–C). Most PHI-base orthologs were found in modules Darkred (*n* = 24) and Purple (*n* = 48) and accounted for 15% and 20% of the total number of genes in these modules. Analyses of KEGG pathway classes (Figure 3D) confirmed that genes associated with metabolism of amino acids (*n* = 33) were highly represented within the three modules, followed by genes associated with metabolism of nucleotides (*n* = 20), complex lipids (*n* = 15), cofactors and vitamins (*n* = 15) or complex carbohydrates (*n* = 11), as well as genes associated with degradation of xenobiotics (*n* = 12). Interactions among genes within modules that were significantly associated with the 96h_S treatment were mapped. Interaction networks for each significant module are shown in Figure 4 for genes associated with KEGG pathway classes, along with hub genes, genes predicted to encode a secreted protein with a signal peptide, the *amtA* gene (OnuG0282), the *hex*1 gene (OnuG6790.1), and two *cyp* genes (OnuG7411 and OnuG7466). The latter four genes were part of the subset of genes that were targeted by knockdown and knockout approaches to investigate their potential roles in fungus growth and pathogenicity.

### 3.4. Functional Analysis of Candidate Pathogenicity Genes

The nine genes selected for functional analysis included: (1) genes that belonged to co-expression modules that were significantly associated with higher expression in S elms (OnuG0282, OnuG6790.1, OnuG7411 and OnuG7466); (2) genes that were more highly expressed in VF elms (OnuG5955); (3) genes that were the most DEG in previous transcriptomic analyses in vitro (OnuG3773); or (4) genes that were part of ongoing studies of genes encoding transcription factors (OnuG1642, OnuG2340) or oxylipins (Onu5504.1). Five of the nine genes had orthologs in PHI-base (Table 3).

RNAi mutants that integrated complete RNAi cassettes (as confirmed by PCR amplification) and demonstrated consistent growth on medium selective for transformants were obtained for the six genes targeted. When grown in liquid OCM containing glycerol as the sole carbon source, the *cox2* mutant did not demonstrate a significant suppression in gene expression compared to WT strain H327 and was therefore not retained for phenotypic assays. Knockout mutants were recovered for the three genes targeted. For each gene knocked down or knocked out, one to three mutants were retained and subjected to preliminary laboratory screening (data not shown), and the individual with the strongest mutant phenotype was retained for further analysis. As shown in Table 3, mycelium of strains cyp570-A4D and mad1-J grew faster than that of their progenitor on MEA, whereas strains hex1-D, Δaox1 5-1 and Δbct2 3-12 grew more slowly. The slow growth phenotype of strain hex1-D was exacerbated in the presence of 0.2 M NaCl, as is typically observed in mutants for this gene. The mycelial growth rate of the cyp570-A4D and cyp52P6-AD mutants decreased significantly in the presence of limonene, as expected. Mutant mad1-J grew significantly more than strain H327 in the presence of NaCl.

Mutants Δaox1-5-1, Δopf2-1-29 and hex1-D were significantly less virulent (*p* < 0.05) than their progenitor on GD apples. However, all mutants were virulent when inoculated to *U. americana* saplings (Table 3). Mutant Δaox1-5-1 induced only 74% defoliation compared to strain Δmus52-1O but the difference was not significant at *p* = 5%.

## 4. Discussion

One hundred years have passed since Schwarz [1] published the results of her observations on the “twig dying of the elms, willows, and peach trees”. Yet, the molecular mechanisms underlying the pathogenic interaction between elms and the pathogens *O. ulmi* and *O. novo-ulmi* remain somewhat of a black box. The work reported herein is part of a larger study that aimed to obtain a genome-wide view of transcriptomic changes associated with interactions between *O. novo-ulmi* and *U. americana*. To this end, one resistant and one susceptible genotypes of American elm were inoculated with yeast spores of the pathogen and sampled 0 and 96h hours later for RNA-Seq analysis of gene transcripts present in elm tissue. Differences in the transcriptomes of the two elm genotypes, as well as major transcriptomic changes induced in the host by inoculation with *O. novo-ulm*i ssp. *americana* strain MH75-4O were reported previously [35]. Recovery of microbial transcripts in sufficient quantity and availability of fully sequenced *O. novo-ulmi* genomes allowed us to investigate the transcriptome of the fungal component of DED.

Overall, the number of fungal sequences found in the four conditions tested is consistent with the fact that the fungus is expected to grow more in the susceptible (S) elm variety than in the resistant (VF) one. Thus, the number of fungal sequences that were recovered from infected trees seems to be proportional to the amount of fungal biomass present. The proportion of expressed *O. novo-ulmi* genes (74%) was lower than in previous RNA-Seq analyses of *O. novo-ulmi* grown in vitro [16,17], probably due to the technical complexity of fungal RNA extraction from infected trees, depth of sequencing effort, and lower fungal concentration in vivo than in vitro. Also, the number of DEGs in the present study is potentially underestimated due to statistical constraints. Nevertheless, our analysis of the *U. americana*-*O. novo-ulmi* interaction transcriptome represents a major departure from previous DED studies. Aoun and collaborators [33] recovered tags for only four fungal genes (compared to 314 elm genes) in their EST analysis of *U. americana* calli inoculated with *O. novo-ulmi* H327. In the first ever RNA-Seq analysis of elms, Perdiguero and colleagues [34] identified 152 genes of DED fungi from *U. minor* inoculated with *O.ulmi*, *O. novo-ulmi* and the fungal endophyte *Daldinia concentrica*. The 6424 fungal genes retained in our analysis thus allowed, for the first time, a truly genome-wide in planta analysis of gene expression in the fungal component of DED.

The transcriptomic comparison of fungi inoculated to trees of contrasted levels of resistance to DED revealed many candidate genes for further pathogenicity experiments. As such, we highlighted a high level of expression of genes encoding HSP proteins in both types of trees, in accordance with the need of response to stressful conditions in the case of host-fungus interactions. In dimorphic fungal human pathogens, HSP-encoding genes have been previously shown to be highly regulated in host-like conditions in *Paracoccidioides brasiliensis*, *Histoplasma capsulatum*, *Cryptococcus neoformans*, and *Candida albicans* [57,58,59,60]. In particular, transcripts of HSP30, HSP70 and HSP90-encoding genes accumulated in *P. brasiliensis* interacting with mouse or human organs. Moreover, HSP90-encoding genes are considered antifungal targets since these genes contribute to fungal viability in species such as *S. cerevisiae* and *S. pombe* [61,62,63]. The identification of *hsp* genes in the set of highly expressed genes in host-interacting conditions in *O. novo-ulmi* suggests similar roles to the ones listed previously in other dimorphic pathogens.

A caveat to the above statement is that HSP-encoding genes were also found to be highly expressed in both yeast and mycelium cultures of *O. novo-ulmi* H327 in vitro [16]. In fact, this was true for over 85% of the 100 genes most expressed in 96h_S and 96h_VF, whether they had PHI-base orthologs or not. Therefore, it appears that exposure to host tissue is not a determinant in the high level of transcription of these genes. On the other hand, the transcription of several genes was clearly stimulated when the fungus was growing in the elm xylem. This was particularly evident in the case of OnuG5955 which had never been detected in transcriptomes of *O. novo-ulmi* H327 yeast cells, mycelium, synnemata or perithecia obtained in vitro [15,16,17,64]. OnuG5955 is orthologous to gene *aox1* encoding an alcohol oxidase belonging to the glucose-methanol-choline (GMC) superfamily of oxidoreductases which make up the AA3 family of auxiliary activities in the CAZy database [65]. Segers et al. [66] showed that Aox1 was a pathogenicity factor in *Cladosporium fulvum*, the aetiological agent of tomato leaf mould, and also reported that carbon starvation conditions induced both transcript induction and protein accumulation by *aox1*. Alcohol oxidase was also suggested to play a role in the modification of terpenoids contributing to plant defense [67]. Targeted deletion of *aox1* in *O. novo-ulmi* H327, however, did not result in a significant loss of virulence on American elm saplings (Table 3). Another gene (OnuG5966) in the same portion of the *O. novo-ulmi* genome was also expressed only in elm. While this putative cell surface protein-coding gene is not included in PHI-base, it may represent another candidate pathogenicity gene worth investigating, along with other candidates among the 90 genes detected only in planta so far.

Among the 100 fungal genes most expressed in *U. americana*, 17 had been previously found to be upregulated in yeast, and only three in 5-d old mycelium in vitro [16]. This observation may reflect the fact that, under our experimental conditions, most of the *O. novo-ulmi* inoculum present in elm xylem vessels at 96 hpi consisted in yeasts, especially in the resistant Valley Forge variety [36].

The plant pathology literature contains several suggestions as to the role of fungal cell-wall degrading enzymes in pathogenesis [68,69] and DED is no exception. The more aggressive *O. novo-ulmi* secretes higher amounts of glycosidases and exo-glycanases than the less aggressive *O. ulmi* in vitro [70,71], whereas production of xylanases was not found to differ between species [72]. Using scanning electron microscopy, Scheffer and Elgersma [73] observed severe erosion of xylem vessels from elms inoculated with *O. novo-ulmi*, whereas vessels from non-inoculated controls looked normal. Our analysis of the *O. novo-ulmi* MH75-4O transcriptome in planta confirmed that genes for many CAZYmes were expressed in elms, usually more in the S than in the VF variety, although the numbers of transcripts generally did not exceed 25. The few genes that were expressed over this threshold encoded mostly glycosyl hydrolases. The discrepancies between our results and those reported previously may, in good part, reflect the different lengths of incubation in elm tissue (96 h in our study versus 4 months in the study by Scheffer and Elgersma [73]).

Likewise, whereas production of laccases was proposed to be associated with virulence and survival of *O. novo-ulmi* within infected trees [74,75], we found no evidence for the expression of five of the six genes encoding laccases (OnuG3850, OnuG6417, OnuG7110, OnuG7274 and OnuG7475), and only minimal expression (<3 transcripts) for OnuG5046. We also inspected our dataset for genes encoding lipases, as they are known to be involved in fungal adhesion and virulence in numerous plant pathogens such as *Alternaria brassicicola, Botrytis cinerea* and *Blumeria graminis* [76,77,78]. Among the 37 lipase-encoding genes for which we detected transcripts, 16 had orthologs in PHI-base. These included two genes (OnuG1496 and OnuG3901) represented by 80 or more transcripts in S elms, as well as two additional genes (OnuG2004 and OnuG7690) that had fewer transcripts but were nevertheless significantly more expressed in S elms and thus are also candidates for further investigation. Additional candidate genes were identified in WGCNA modules Darkred, Green and Purple which included several genes involved in the metabolism of lipids and complex lipids.

Toxins play an important role in several plant-fungus pathosystems and, not surprisingly, were suggested to be involved in the etiology of DED. Although *O. novo-ulmi* produces phytotoxic molecules including phenolic compounds [79] and glycopeptides [80], there is so far no experimental evidence that these molecules contribute to pathogenicity or virulence. Nevertheless, Comeau and collaborators [11] annotated a total of 48 genes encoding cytochrome P450s in the *O. novo-ulmi* H327 genome and suggested that some of these genes might enable *O. novo-ulmi* to produce sterigmatocystin (ST), a precursor of aflatoxin. The latter is a highly toxic, carcinogenic secondary metabolite found in *Aspergillus* species [81,82] and ST is present in multiple fungal plant pathogens to which it confers advantage during host infection [83]. In *Aspergillus parasiticus*, the final steps of aflatoxin production involve two enzymes, a CYP450 (CYP64 family) further characterized as a O-methylsterigmatocystin (OMST) oxidoreductase, and the ST 8-O-methyltransferase (encoded by genes *aflQ* and *alfP*, respectively) [81,84]. Genes encoding six OMST oxidoreductases and two ST methyltransferases occur in the *O. novo-ulmi* genome [11]. One gene for each enzyme was found in the set of genes overexpressed 96h post-inoculation in the S elm variety. In addition, *O. novo-ulmi* possesses orthologs of genes controlling HC toxin production in the plant pathogen *Cochliobolus carbonum*. This secreted molecule is essential for disease in maize cultivars that have the Hm resistance gene [85]. Again, the upregulation of the *O. novo-ulmi* gene (OnuG8024) annotated as the HC-toxin efflux carrier TOXA in the S elm variety and the downregulation of a TOXD gene (OnuG7311) in the VF variety suggest a role for these two genes preferentially in the susceptible elm variety, thus linked to pathogenicity in *O. novo-ulmi*. It is noteworthy that OnuG7311 was never found to be expressed in previous transcriptomic analyses of strain H327 in vitro [15,16,17,64]. Comparative analysis of *O. novo-ulmi* wild-type and mutants in which the above genes are targeted would determine whether in silico results are also meaningful in planta.

Similar to CYP450s, polyketide synthases are enzymes involved in polyketide production. Polyketides are secondary metabolites that include a large variety of antibiotics or mycotoxins [85] such as the aflatoxin mentioned previously. The genome of *O. novo-ulmi* strain H327 contains only ten genes coding for PKSs. These genes were identified as potential backbone genes for clusters responsible for the production of secondary metabolites [55]. Interestingly, *pks8* (OnuG7312) is only present in *O. novo-ulmi* and *O. ulmi* compared to other Ophiostomataceae species, and therefore may be specific to DED fungi [55]. Gene *pks7* (OnuG6972) was the backbone/cluster gene the most expressed in planta. It was also strongly differentially expressed, with 55 times more transcripts recovered in the S than in the VF elm variety. This gene is specific to *Ophiostoma* and *Sporothrix* species but its function remains known [55]. The fact that all of these *cyp450* and *pks* genes were preferentially expressed when *O. novo-ulmi* interacts with the S elm variety might reflect the activation of a pathway that leads to the production of an aflatoxin-like compound contributing to pathogenicity. Further metabolic investigations are needed to find out if *Ophiostoma* species produce aflatoxin and if this molecule interacts with living elm tissue.

In the *O. novo-ulmi* H327 genome, 16 CYP450-coding genes were identified as being potentially involved in the detoxification of plant defense compounds [11]. Out of this set, we highlighted genes encoding two benzoate 4-monooxygenases (OnuG3693 and OnuG7958) and one pisatin demethylase (OnuG7411) preferentially expressed in the S elm variety. Although knockdown of gene OnuG7411 by RNAi resulted in lower tolerance to limonene in vitro, it did not affect the virulence of the mutant towards GD apples and *U. americana* saplings. Phytoalexins are antimicrobial compounds produced by plants in response to infection [86,87,88]. Previous in vitro and in planta studies of phytoalexins produced by the American elm in response to *O. novo-ulmi* infection showed that the major phytoalexins include (iso)flavonoids, phenolic compounds, as well as mansonones which are sesquiterpene quinones [89,90]. Production of mansonone-like phytoalexins in elm trees is induced by a glycoprotein [91] encoded by OnuG1041 which, however, was barely expressed in our dataset. Another CYP450 gene found in the *O. novo-ulmi* genome (OnuG7466 encoding a CYP52P6) is linked to modifications of plant defense terpenes and was up-regulated in the 96h_S treatment and included in the Purple WGCNA module. Gene OnuG7466 is thought to be involved in the transformation of α-pinene to verbenol which is an aggregation pheromone for bark beetles, therefore attracting them to the diseased tree, a phenomenon that favors fungal dissemination [11,92,93]. Elm trees are known to produce terpenes (β-pinene, α-cubebene, spiroaxa-5,7-diene, limonene and δ-cadinene) and fungal infection stimulates the production of these compounds [94]. A previous study on the response of *O. novo-ulmi* to terpenes showed no conclusive impact of terpenes on expression of OnuG7466 [11]. The contrasting result here suggests that this earlier observation was probably due to the choice of the reference gene for the qRT-PCR assay (OnuG2373, CYP51F1) which was later reported to be differentially expressed in *Leptographium qinlingensis* in response to terpenes [95]. When transcription of OnuG7466 was knocked down by RNAi, the resulting mutant was found to be more sensitive than WT strain H327 to limonene in vitro but remained virulent on both GD apples and *U. americana* saplings. Nevertheless, identification of differential expression when the fungus was inoculated to S and VF elm varieties suggests that terpenes might be produced differentially in these two varieties and that OnuG7466 is responding to this difference.

The role of cerato-ulmin (CU) in the etiology of DED is still controversial. Originally considered as a wilt toxin contributing to *O. novo-ulmi* virulence [19], CU was later reclassified as a hydrophobin [96]. Mutants with a disrupted CU gene were found to be as virulent as the wild- type *O. novo-ulmi* [21], whereas some transformants of the saprobe *O. quercus* overexpressing the *O. novo-ulmi* CU gene induced wilting on elms [97]. We verified if the level of elm resistance to DED influenced CU gene expression, as previously suggested by Sherif and collaborators [36] based on qRT-PCR and reporter gene assays. We found the same result, as the *CU* gene (OnuG4296) was more highly expressed in the S elm variety versus the VF elm variety [36]. Whether this reflects increased fungal biomass production or points to a role in pathogenesis is unknown.

Analysis of the fungal genes that were the most differentially expressed between the 96h_S and 96h_VF treatments highlighted an overrepresentation of genes encoding transporters of nitrogenous compounds. These included *mep2*, which encodes the permease with the highest affinity for ammonium/methylamine involved in the dimorphic switch in *S. cerevisiae*, *C. albicans* and *U. maydis* [98,99,100]. Given the high level of differential expression of the *O. novo-ulmi mep2* ortholog (OnuG1681) at 96h in the S elm variety, this gene could reflect the link between dimorphic switch and pathogenicity in *O. novo-ulmi*. Based on positional cloning (unpublished results) and annotation results, another gene (OnuG0282) encoding the AmtA ammonium transporter may actually be the *pat1* locus described by Et-Touil et al. [23,101]. As we noted previously for lipase-encoding genes, OnuG0282 produced few transcripts at 96 hpi but its differential expression in S and VF elms made it a good candidate for further studies. An RNAi knockdown mutant was obtained and its phenotype assayed in vitro and in planta. The mutant, however, exhibited a behavior that was similar to that of WT strain H327.

Inoculation of resistant trees induces a higher level of stressful conditions for the fungus. For instance, Islam et al. [35] recently reported that defense response (GO:0006952) was among the topmost significantly enriched GO terms in VF elms in response to *O. novo-ulmi*. Therefore, it is not surprising that, in the pathogen, genes involved in stress response such as CAZyme-encoding genes and genes responding to oxidoreduction stresses were overrepresented in the portion of overexpressed genes in 96h_VF vs. 96h_S. This suggests that living conditions in more resistant elm varieties tree induce an increased consumption of carbohydrate compounds for respiration and survival.

In a previous investigation of transcriptomic changes associated with *O. novo-ulmi* yeast-mycelium transition in vitro, the top DEG was gene OnuG3773 whose level of expression increased almost 8000 times within 27 h of incubation [17]. The expected product of OnuG3773 is an extracellular serine-threonine-rich protein related to the Adhesin protein Mad1 reported to contribute to virulence of the fungus *Metarhizium anisopliae* to insects [102]. We hypothesized that since OnuG3773 was overexpressed in mycelium, it might be involved in interactions with elms rather than with bark beetle vectors. Gene OnuG6790.1, which is an ortholog of the gene coding for a major protein (Hex1) in Woronin bodies, was another highly upregulated gene during yeast-mycelium transition in vitro (959-fold increase within 27 h of incubation) [17]. This gene was also highly expressed both in S and VF elms (390 and 97 transcripts, respectively). Deletion of *hex1* was reported to alter production of haustoria and decrease pathogenicity in the rice blast fungus *Magnaporthe grisea* [103] and inhibit the production of adhesive traps in the nematophagous fungus *Arthrobotrys oligospora* [104]. Therefore, genes OnuG3773 and OnuG6790.1 seemed good candidates for functional validation assays. The virulence of knockdown mutants Mad1-J and Hex1-D, however, was found to be comparable to that of WT strain H327 on *U. americana* saplings. Although the genes targeted in these mutants were not totally silenced, levels of residual expression were low (2.8% and 6.5% for *mad1* and *hex1*, respectively) and it therefore seems unlikely that null mutants would behave differently in elm tissues. Further analysis of *O. novo-ulmi* H327 genome annotations led us to identify a putative ortholog (OnuG1509) of the *M. anisopliae mad2* gene encoding another adhesin which is required for colonization of plants by the fungus [102]. Although OnuG1509 was expressed in low levels in planta (18 and 2 transcripts detected in S and VF elms, respectively), work in under way to verify whether this gene plays a significant role in the DED pathosystem (de Oliveira, unpubl. data).

Since genetic transformation systems for *O. novo-ulmi* were developed [105], the contribution of 13 genes to pathogenicity (including eight genes analyzed in this work) has been investigated by producing and phenotyping knockdown- and knockout mutants [21,22,30]. Genes OnuG6779 (*ade7*) and OnuG7434 (*ade1*), which code for two enzymes involved in the adenine biosynthesis pathway, were shown to be required for pathogenicity [30], thereby confirming previous reports of altered pathogenicity phenotype in adenine auxotrophs from several plant pathogenic fungi (e.g., [106]). Mutants for the other candidate genes were all highly virulent. Since seven out of these eleven genes (64%) had orthologs in PHI-base and/or were DEG in planta, this suggests that in silico annotations and transcriptomic data are not good predictors of the contribution of genes to pathogenicity and virulence in *O. novo-ulmi*. This result, however, must be put into context since several factors may lower chances of identifying mutants with significantly lower virulence. First, the set of genes analyzed so far is very small. Second, American elm is highly sensitive to DED and it is likely that some mutants with altered virulence are still able to successfully colonize this species but would fail to do so on a moderately resistant species such as *U. procera*. For instance, *O. novo-ulmi* strains carrying the *pat1-m* allele were significantly less virulent than *pat1-h* strains when inoculated to *U. procera* [23] but both strain genotypes were highly virulent on *U. americana* [52]. Third, variation within treatments was evident in pathogenicity tests on *U. americana* conducted in this work and this may have contributed to concluding that mutant ΔAox1-5-1 was not less virulent than its Δmus52 progenitor. Future tests should include clonal material, such as the S genotype described by Islam et al. [35]. Fourth, the basal inoculation procedure used for testing mutants puts elm saplings under intense pathogen pressure [107]. Finally, many of the genes targeted so far may be moderate contributors to virulence and therefore mutations in individual genes go undetected in pathogenicity tests. This hypothesis is being tested by producing double- and triple mutants obtained by controlled sexual crosses between single mutants (de Oliveira, unpubl. data). Therefore, using optimized inoculation protocols and testing mutants representing a larger set of candidate genes (including gene combinations) may soon lead to the formal identification of genes that contribute to pathogenicity and virulence of *O. novo-ulmi* towards *U. americana* and other elm species.

## 5. Conclusions

The present study is the most comprehensive genome-wide analysis of gene expression in *O. novo-ulmi* infecting elms. Together with the work recently reported by Islam et al. [35] based on the same experimental setup, it provides novel information on the transcriptomics of interactions between this highly aggressive pathogen and elm genotypes with contrasted levels of resistance, namely a resistant (Valley Forge) and a susceptible (S) varieties of *U. americana*. Although the proportion of *O. novo-ulmi* genes expressed in planta (74%) was lower than that reported in previous RNA-Seq experiments that were conducted on material grown in vitro (from 88% to 90%; [16,17]), we nevertheless identified a rich repertoire of fungal genes, including close to 500 genes that were differentially expressed between VF and S elms at 96hpi. We also unveiled networks of fungal genes that were co-expressed during the infection of S elms. Targeted knockdown or deletion mutants obtained for the eight genes we assessed initially did not show significant changes in virulence but we expect that the combination of transcriptomic data and tools for efficient production of mutants [30] will facilitate the formal identification of genes controlling pathogenicity and virulence in *O. novo-ulmi*. In turn, this may help breeders identify elm genes encoding molecules and receptors that interact with fungal factors in order to select and breed American elms that are more resistant to DED.

## Figures and Tables

**Figure 1 jof-08-00637-f001:**
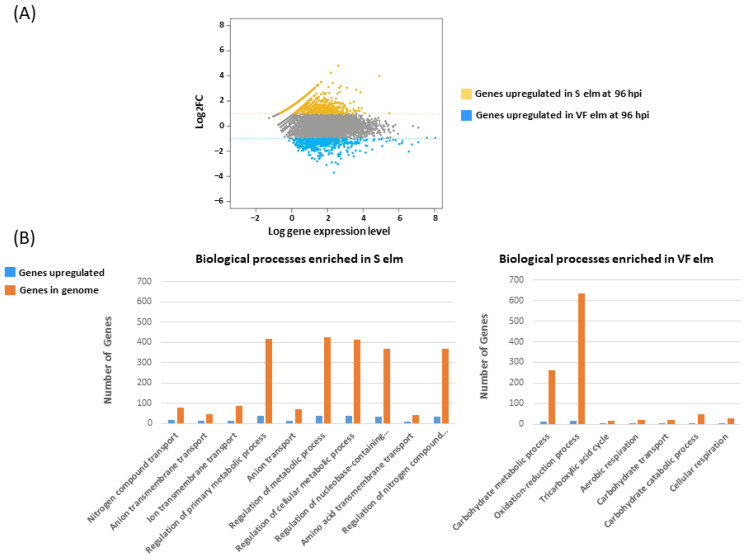
Differentially expressed *Ophiostoma novo-ulmi* genes during interaction with *Ulmus americana*. (**A**) MA plot of the 6424 *O. novo-ulmi* genes that were detected at 96 hpi showing genes that were overexpressed in susceptible (S) and resistant (VF) elm genotypes. Read counts were normalized using DESeq2 in R [56] with log_2_FC > 1 and scatter plot visualization. (**B**) Gene ontology (GO) terms for biological processes that were significantly enriched in *O. novo-ulmi* colonizing susceptible (S) or resistant (VF) elm genotypes. The top 10 processes are shown in the case of interactions with S elm (out of 39 GO terms for biological processes), whereas all enriched terms for processes are shown in the case of interactions with VF elm.

**Figure 2 jof-08-00637-f002:**
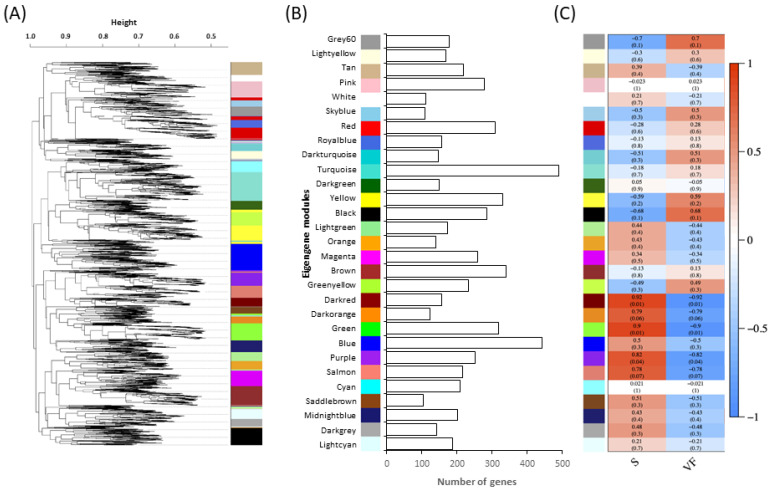
Weighted gene co-expression network analysis (WGCNA) of the fungal component of the *Ulmus americana*-*Ophiostoma novo-ulmi* ssp. *Americana* interactome. (**A**) Cluster dendrogram showing the genes (branches) and co-expressed modules (colors); genes were clustered in 29 modules according to 1-TOM soft threshold. (**B**) Number of eigengenes per module. (**C**) Module traits in *O. novo-ulmi* in planta. The colour scale (red-blue) for relationships between module eigengenes (rows) and treatments (columns) represents the strength of the correlation (1 to −1). Modules Darkred (*p* < 0.01), Green (*p* < 0.01) and Purple (*p* < 0.05) were associated with higher gene expression in susceptible (S) compared to resistant (VF) elm.

**Figure 3 jof-08-00637-f003:**
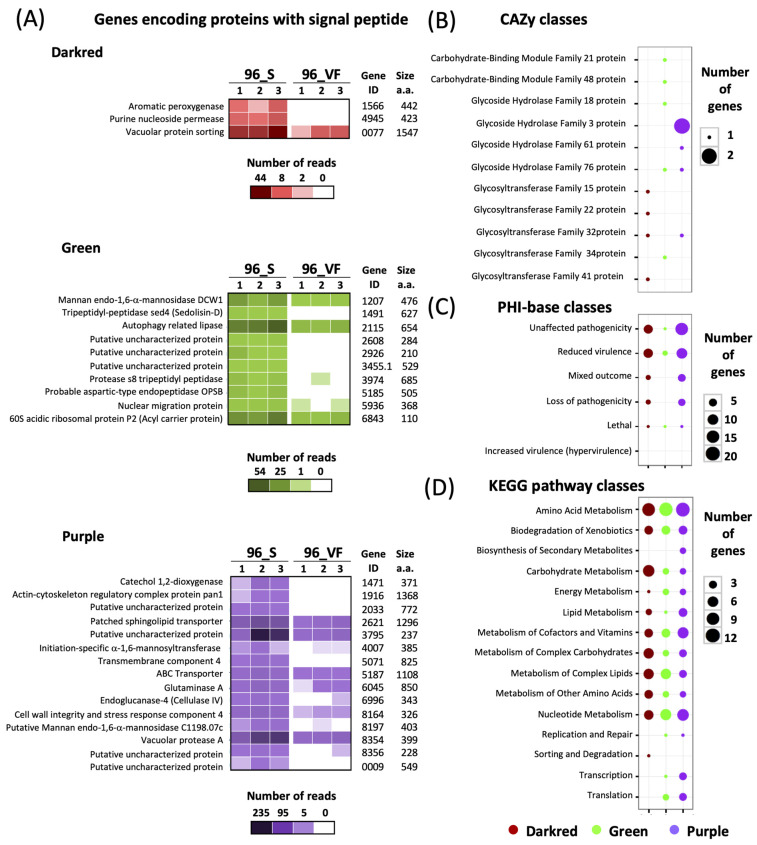
Annotation of genes found in weighted gene co-expression network analysis modules Darkred, Green and Purple that were associated with higher *Ohiostoma novo-ulmi* ssp. *Americana* gene expression in susceptible *Ulmus americana*. (**A**) Identity and expression level of genes encoding a protein with a signal peptide. Number of reads is shown for each replicate. (**B**) CAZY classes. (**C**) PHI-base classes. (**D**) Kyoto Encyclopedia of Genes and Genomes (KEGG) pathway classes.

**Figure 4 jof-08-00637-f004:**
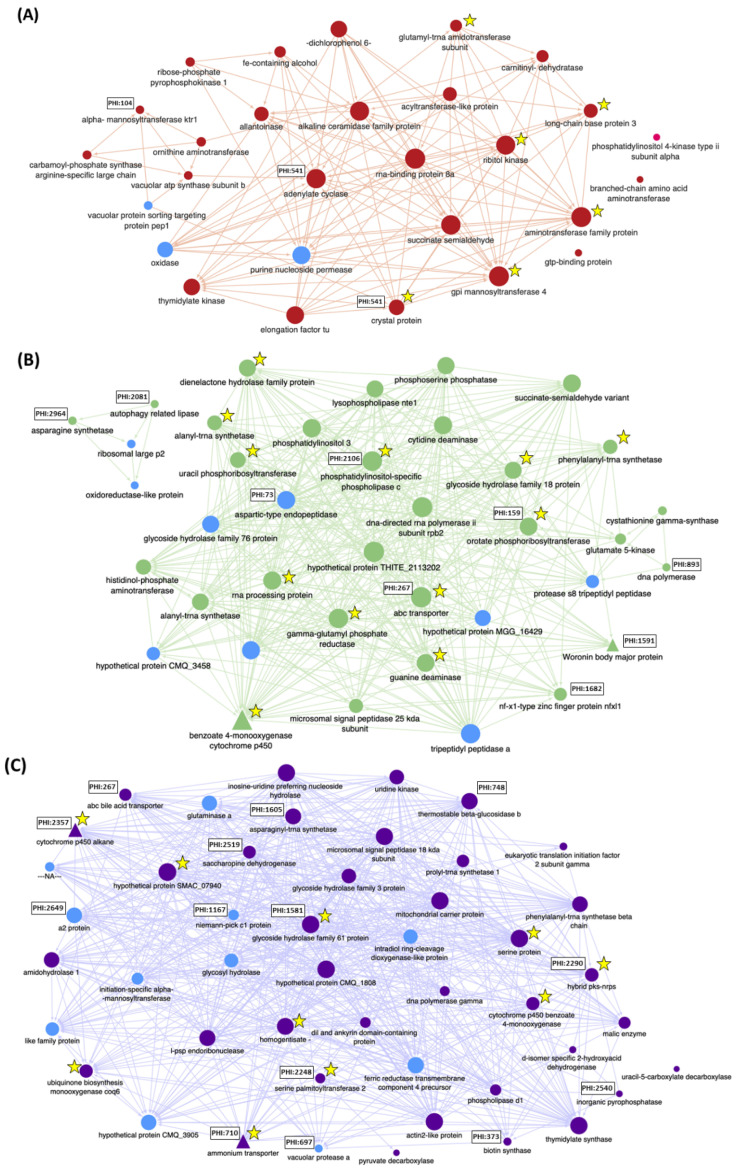
Interaction networks for selected genes in weighted-gene co-expression network analysis modules Darkred (**A**), Green (**B**) and Purple (**C**) that were associated with higher *Ohiostoma novo-ulmi* ssp. *americana* gene expression in susceptible (S) *Ulmus americana*. Interactions are shown for genes associated with KEGG pathway classes, along with genes predicted to encode a secreted protein with a signal peptide (light blue), and genes for which knockdown- or knockout mutants (triangles) were obtained and tested. Genes upregulated in S elms are identified by a star. PHI-base identifier is shown for genes with orthologs in the PHI-base curated database of genes involved in host-pathogen interactions. The size of the gene symbol (circle or triangle) is proportional to the number of interactions with other genes in the module.

**Table 1 jof-08-00637-t001:** Overview of RNA-Seq data for *Ulmus americana*-*Ophiostoma novo-ulmi* ssp. *americana* interaction.

	Valley Forge		Susceptible	
	0 h	96 h	0 h	96 h
**Total reads ^1^**	75,742,361	73,144,879	66,441,814	75,518,322
**Reads fully mapped to *O. novo-ulmi* genes (exons + introns)**	0	24,193	0	65,910

^1^ Data from [35].

**Table 2 jof-08-00637-t002:** Summary of categories in *Ophiostoma novo-ulmi* ssp. *americana* differentially expressed genes.

Treatment	Expressed	Upregulated (logFC(1) ^1^)	Annotated Upregulated	CAZYmes Upregulated	Signal Peptides Upregulated	PHI-Base Upregulated
96h_S	5392	370	303	16	20	94
96h_VF	3014	102	84	17	26	29

^1^ Log_2_ of the Fold Change.

**Table 3 jof-08-00637-t003:** Phenotypes of knockdown- and knockout mutants of *Ophiostoma novo-ulmi* in vitro and in planta.

Strain	Gene	Annotation	PHI: Base	Method ^1^	%RGE ^2^	Nb Transcripts ^3^		DEG ^4^	Module ^5^	Mycelial Gowth ^6^			Virulence ^7^	
						96_S	96_VF			MEA	NaCl	Lim	Apple	Elm
Cyp570-A4D	7411	Pisatin demethylase	−	RNAi	25.3	5.0	0.0	Up96_S	Gre	131.3 ^8^	n.t. ^9^	81.0 *	97.0	96.3
Cyp52P6-AD	7466	Putative α-Pinene to Verbenol enzyme	−	RNAi	3.4	17.3	1.0	Up96_S	Pur	117.8	n.t.	69.0 *	105.0	87.5
Hex1-D	6790.1	Hex1 Woronin body formation	1591	RNAi	6.5	389.7	97.0	No	Gre	78.0 *	45.2 *	53.7 *	55.0 *	100.6
Mad1-J	3773	Mad1 Adhesin	−	RNAi	2.8	10.3	4.3	No	Mag	128.5 *	145.2 *	108.9	84.0	101.0
AmtA-B	282	Ammonium transporter	2710	RNAi	1.6	13.3	0.0	Up96_S	Pur	109.2	n.t.	n.t.	109.0	101.0
ΔAox1 5-1	5955	Aox1 Alcohol oxidase	199	OSCAR	n.t.	567.3	808.0	Up96_VF	GrY	84.9 *	n.t.	n.t.	62.2 *	74.2
ΔOpf2 1-29	1642	Opf2 Transcription Factor	1931	OSCAR	n.t.	33.0	16.0	No	GrY	100.7	n.t.	n.t.	69.4 *	100.1
ΔBct2 3-12	2340	Bct2 Transcription Factor	1933	OSCAR	n.t.	20.3	4.0	No	Ora	81.6 *	n.t.	n.t.	96.9	100.2

^1^ Knockdown (RNAi) mutants were derived from *O. novo-ulmi* ssp. *novo-ulmi* H327, whereas knockout (OSCAR) mutants were obtained from strain *O. novo-ulmi* ssp. *novo-ulmi* Δmus52-1O. Both strains were highly virulent on Golden Delicious (GD) apples and *Ulmus americana* saplings. ^2^ % Residual gene expression measured in knockdown mutants compared to H327 progenitor. ^3^ *Ophiostoma novo-ulmi* ssp. *americana* MH74-4O transcripts detected in susceptible (96_S) or resistant (96_VF) saplings of *Ulmus americana* at 96 h post-inoculation. ^4^ Differentially expressed genes (DEGs) were significantly more expressed either in susceptible elms (Up96_S elm) or in VF elms (Up96_VF elm). ^5^ Modules determined by Weighted Gene Co-expression Newtork Analysis (WGCNA): Gre: Green; Pur: Purple; Mag: Magenta; GrY: Green Yellow; Ora: Orange. ^6^ MEA: Malt extract agar; NaCl: MEA supplemented with 0.2 M NaCl; Lim: MEA supplemented with 1:1000 limonene. ^7^ Virulence on GD apples was assessed by measuring the diameter of necrosis after 4 weeks. Virulence on elms was assessed by mesuring the % defoliation of American elm saplings after 3 weeks. ^8^ Data for mycelial growth, and virulence on GD apples and *U. americana* saplings are presented as % of values recorded for progenitor strains H327 (for RNAi mutants) or Δmus52-1O (for OSCAR mutants). ^9^ n.t.: not tested. * indicates significant difference (*p* < 0.05) between mutant and progenitor strain.

## Data Availability

RNA-Seq experimental data corresponding to *U. americana* and *O. novo-ulmi* ssp. *americana* raw reads for each biological replicate were deposited at the NCBI Sequence Read Archive as Binary Alignment Files under the accession number SRP149721. The assembled transcriptome of *U*. *americana* is also available at NCBI SRA library as a FASTA file under the same accession number.

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
