# Peer review of "Comparative Analysis of Transcriptomes of Ophiostoma novo-ulmi ssp. americana Colonizing Resistant or Sensitive Genotypes of American Elm"

_jof, 2022, doi:10.3390/jof8060637_

Round 1

Reviewer 1 Report

O. novo-ulmi is a key pathogen causing the continuous epidemic of DED and has strong pathogenicity. The underlying molecular mechanisms of its pathogenicity and virulence remain largely unclear. This paper carried out the transcriptome study of O. novo-ulmi resistant and sensitive Ulmus americana saplings and screened O. novo-ulmi pathogenic genes by loss-of-function approaches. It is to be helpful to breed new resistant elm trees and reduce the occurrence of DED. However, there are certain problems in the expression and writing of the article, which are suggested to be revised, as follows.

Introduction

Progressive logical relationship need to be strengthened in the introduction, especially in the fourth paragraph of introduction, for example, description of research status on O. novo-ulmi ssp. americana pathogenicity, the research advances of DED in two types of American elm, and screening basis of candidate pathogenic genes in other types of plant pathogens.

line 46-48:

The above part is about the harm and evolution of O. novo-ulmi's two subspecies, the third species is not necessary to mention here.

line 58-66:

What’s the point of description of the DED fungi dimorphic growth here? It seems unnecessary.

line 87-88:

Extend examples of how RNAi technology can efficiently generate targeted mutants

line 97-99:

After the sentence, there should be something added, to explain or show the progress in literature how the development of dual RNA-sequencing technology has improved the sensitivity of detection.

Materials and Methods: suggest to briefly explain the reason why the article selected amtA, aox1 and other genes as candidate pathogenic genes to validate their function, because the introduction stated that there is no experimental evidence for any of DED fungal candidate pathogenic genes,

 2.4. Functional Analysis of Candidate Pathogenicity Genes:

Supplementary Tables 1 and 2 do not have amtA but with Mep2 gene RNAi cassettes nucleic acid sequences and qPCR amplification primers

line 259-260:

Specific media were used for the cultivation of some mutants, but they are not consistent with “mycelium growth” column in Table 3, for example for hex1 and Mad1-J mutants, are MEA supplemented with NaCl and limonene used for two mutants?

Results:

 line 277-280:

The description is wordy.

line 295-297:

Need to state the meaning of determining the correlation of gene expression levels between two treatments? Does it indicate gene expression levels are associated with susceptibility?

line 359-361:

In which elm is this gene up-regulated?

line 463:

There is no OnuG6990.1 gene in Table 3.

line 467:

eight or nine?

line 473-475:

Put this part in the method.

Line 483:

Should abpf2-1-29 be Opf2 1-29?

line 482:

In Table 3, mycelial growth of mad1-J mutants in NaCI should have an asterisk.

Discussion:

Suggest to divide the discussion into section with several titles. The purpose of this study was to screen the pathogenic genes through resistant differentiation of two elm species. What is the basis and principles for pathogenic gene selection? Explain why multiple pathogenic related genes, which are also overexpressed or differentially expressed (in lines 595-600, 620-623, and 631-634), are not selected?

line 539-547:

Is a highly expressed gene a pathogenic gene? Some examples of plant-host interactions should be found to illustrate the selection basis for candidate genes

Line 565:

eletion should be deletion.

Line 644:

Should OnuG7598 be OnuG7958?

Reviewer 2 Report

Dear Authors,

Do the present findings bring us closer to controlling elm resistance? We better understand the process of elm resistance at the molecular level, which involves the expression of HSP genes. Will it be possible in the future to create genetically modified trees that are more resistant to DED? What practical application do the authors have in mind for future research results?

Elm breeding programmes have selected resistant elm cultivars currently used for afforestation [Martín, J. A., Solla, A., Oszako, T., & Gil, L. 2021. Characterizing offspring of Dutch elm disease-resistant trees (Ulmus minor Mill.). Forestry: An International Journal of Forest Research, 94(3), 374-385]. So far, selected disease-resistant elms have narrower wood vessels than susceptible trees (although this is not conclusive), but it is likely that this characteristic physically limits infection (or its spread). Are the genes that may influence this trait known, or has their expression perhaps been observed?

How might climate change (global warming) affect resistance of elms to DED disease?

Could O. novo-ulmi be more aggressive in an atmosphere with higher CO2 concentrations? I am thinking of stronger tree growth and wider wood vessels that might facilitate spore penetration, germination, and mycelial development in the xylem?

Why do fungi produce toxins when they have not been shown to be involved in pathogenesis? What further studies should be done to find out? What is the role of phytoalexins if studies have not found expression of genes responsible for their production?

Can detection of terpenes (or other volatile compounds) help foresters identify weakened trees that are susceptible to infection? Electronic noses and/or trained dogs that can detect harmful insects and plant pathogens are being developed.

Can crossing less virulent (pat1-m) strains with virulent (pat1-h) strains reduce U. procera infections?
